# Long-Term Compressive Strength of Polymer Concrete-like Composites with Various Fillers

**DOI:** 10.3390/ma13051207

**Published:** 2020-03-07

**Authors:** Joanna Julia Sokołowska

**Affiliations:** Department of Building Materials, Faculty of Civil Engineering, Warsaw University of Technology, 00-637 Warsaw, Poland; j.sokolowska@il.pw.edu.pl; Tel.: +48-22-2346482

**Keywords:** long-term compressive strength, durability, polymer composites, polymer concrete, mineral fillers, fossil fuel combustion by-products, fly ash

## Abstract

The durability of building composites with polymer matrix, such as polymer concretes, is considered high or excellent. However, very few studies are available that show the properties of such composites tested long after the specimens’ preparation, especially composites with fillers other than traditional rock aggregates. The paper presents the long-term compressive strength of polymer concrete containing common and alternative fine fillers, including quartz powder (ground sand) and by-products of the combustion of Polish fossil fuels (coal and lignite), tested nine or 9.5 years after preparation. The results were compiled with the data for respective specimens tested after 14 days, as well as 1.5 and 7 years. Data analysis confirmed the excellent durability of concrete-like composites with various fillers in terms of compressive strength. Density measurements of selected composites showed that the increase in strength was accompanied by an increase in volumetric density. This showed that the opinion that the development of the strength of composites with polymer matrices taking place within a few to several days was not always justified. In the case of a group of tested concrete-like composites with vinyl-ester matrices saturated with fly ashes of various origins, there was a further significant increase in strength over time.

## 1. Introduction

Polymer concrete-like composites are similar to ordinary concretes, but with completely cement-free polymer matrices or matrices of two co-binders, i.e., mineral cement and a significant amount of polymer. The examples of such composites are polymer concretes (PC) and mortars with fillers of various sizes and polymer-cement concretes (PCC) with complementing polymer-cement mortars and pastes [1,2,3,4].

Each time, presenting the concept of polymer concrete-like composites and their advantages to researchers familiar only with traditional concrete (that is often slightly modified with polymers, e.g., through the use of polymer-based admixtures), at some point, high durability [1,2,3,4] was indicated. The high or excellent durability of these composites is mainly the consequence of the presence of a significant amount of polymer, which allows obtaining tighter matrices. Therefore, such matrices are more resistant to external environmental onslaughts, including destructive chemical processes (acid and alkaline corrosion, leaching of easily soluble compounds, etc.) [5,6,7,8], as well as physical processes (freezing and thawing of penetrating water, thermal shock or, in the case of PCC, destruction resulting from increasing the volume of crystallization products, etc.) [2,9,10,11]. At the same time, the matrices present higher adhesion to fillers. As a result, composites with such matrices are mechanically stronger [2,6]. However, not much of the published data concerned polymer concrete-like composites tested at a later age, confirming their ability to maintain mechanical properties over time, i.e., confirming their durability due to this criterion. The issue of durability requires even more clarification when wastes or by-products are applied to the aforementioned composites as the substitutes of the regular components. Such composites have been recently eagerly designed and produced, following the assumptions of the sustainable development concept, i.e., a concept strongly promoted for the past few years. Composites containing such wastes and/or by-products (e.g., fly ash, perlite powder, recycled glass, aggregate leftover mineral dust, etc.) are subjected to various tests [5,6,7,9,10,11,12,13], but usually only at the early stage of their service-life. The level of mechanical properties of such composites after a long period of use remains unclear.

The aim of the paper is to discuss the “long-term compressive strength” of polymer concretes with traditional quartz fillers (including coarse and fine aggregates, the same as those used in ordinary Portland cement concrete and very fine quartz powders) and sustainable secondary materials, namely the by-products of the combustion of two kinds of fossil fuels, i.e., coal (or so-called “hard coal”) and lignite (or “brown coal”), mined in deposits located in Poland. The combustion processes are intended to produce electricity or heat. Due to differences in the construction of furnaces and associated combustion installations, the remaining fly ashes differ in terms of morphology and granulation, so they differently affect the consistency of PC mix [13] which, among others, influences the microstructure and, therefore, the properties of hardened composites containing such fly ashes.

The tested composites are cement-free (the binder consists solely of vinyl-ester resin). There is also no water in the composition of these composites. Water disturbs or prevents setting of the vinyl-ester resin (to avoid the negative effect of moisture from the aggregate on the binder setting, aggregates are usually dried before dispensing them into the mix). In the case of this type of concrete, there is no hydration reaction, and the fly ashes present in the composite, despite the fact that they contain amorphous silica, do not participate in the pozzolanic reaction. Therefore, fly ashes in PC are only a very fine filler/aggregate fraction.

The influence of the presence of siliceous fly ash (the by-product of conventional coal combustion) on the properties of polymer mortars and concretes is relatively well recognized. Such composites may present better mechanical properties and workability [5,6,14,15], which is partly due to the almost perfectly spherical shape of the siliceous fly ash particles. The effect of the presence of by-products of coal FBC (fluidized bed combustion), their granulation, and morphology on the properties of PC is less recognized. The fly ash remaining after hard coal fluidized combustion consists of much finer irregular particles, and its chemical composition contains more calcium compounds (up to over 20% by mass). The fly ash remaining after lignite fluidized combustion is a mixture of spherical and irregular particles and contains less calcium compounds (but still more than siliceous fly ash). It can be found that the use of microfillers of an increased content of calcium compounds as substitutes for quartz fillers may lead to a greater increase in the compressive and flexural strength of polymer composites [16].

## 2. Genesis of Research

The results presented in the paper were obtained by the continuation of research conducted in 2010–2011 [7,17,18] that was focused on assessing the possibility of using various fly ashes as substitutes for regular microfillers (i.e., fillers of a size usually up to 120 µm [19]) in polymer composites, and later research on the durability of such composites conducted in 2018 [20]. In the framework of the second study, the author carried out a series of mechanical tests of vinyl-ester concretes 1.5 and seven years after production. The results of tests carried out on 1.5-year-old specimens showed a clear increase in compressive strength in comparison to the strength of 14-day-old specimens. The increase ranged from 8.6% to 36.2% (on average, by 18.4%). The tests carried out on seven-year-old specimens also showed a noticeable increase in compressive strength. The upward trend was observed in the case of polymer composites containing both types of fly ash: siliceous fly ash (the by-product of conventional coal combustion) and FBC fly ash. In the case of the composite with fluidized fly ash, strength gain values were even higher (a maximum of 60.4%, 22% on average). The conclusion from that research was that for polymer concretes containing fly ash being a coal combustion product (CCP), the development of compressive strength occurs for a long time [20].

Such a conclusion contradicts the opinion that the development of the strength of composites with polymer matrices takes place within a relatively short time, or at least indicates that it is not the correct opinion in every case. Figure 1a presents the general concept of the development of compressive strength in time of two concrete-like composites, i.e., polymer-cement concrete (PCC) (of a matrix that is partly cementitious and partly polymeric) and polymer concrete (PC) (only polymer matrix). The concept assumes that the development of polymer concrete compressive strength is very intense and short, i.e., PC develops 60% of the maximum strength within one day and ca. 80% of the maximum strength within three days [2]. This is related to the dynamics of the setting and curing of the polymer binder, usually the thermosetting resin, which quickly goes through the phase of gelation to the more and more intense polymerization and creation of the 3D cross-linked structure of the polymer matrix. Thermosetting resins tend to develop the vast majority of the mechanical strength within first days (e.g., epoxy resins develop about 80% of the vast majority of the strength within the first 1–2 days) [21]. This is why polymer concretes tend to develop utility efficiency after several days (which depends on humidity and ambient temperature; the abovementioned model assumed temperature no lower than 15°C). Generally, after 14 days, further significant development of mechanical strength is not expected.

Figure 1b presents a trial of recalculation of the model using the natural logarithm function that allowed indicating the moment of achieving potentially full (100%) compressive strength as 52 days.

The empirical data obtained in the framework of the previously mentioned research [20] compared to the abovementioned model (determined for polymer concrete with an assumed maximum compressive strength of 100 MPa) with the time variable expressed on a logarithmic scale are given in Figure 2. In the case of a group of vinyl-ester composites containing various fly ashes, there was a clear tendency for further development of compressive strength.

The newly obtained data presented in this paper, i.e., compressive strength of 9.5-year-old concretes, partly complemented earlier findings, as concretes identical in terms of quantitative and qualitative composition were tested at a later date to see if there was even a further increase in strength. What is more, the base of the composites analyzed in terms of long-term compressive strength (considered as a kind of measure of durability) was supplemented with vinyl-ester concretes containing fly ashes from lignite combustion—tested 14 days and nine years after production.

## 3. Materials and Methods

The polymer used to prepare all concretes presented in the study was synthetic vinyl-ester resin of low viscosity (350 ± 50 mPa·s at 25 °C) and high flexural strength and tensile strength (declared by the producer as respectively 110 MPa and 75 MPa). Therefore, concretes made from this resin should retain the high mechanical strength in long-term exploitation, even when exposed to aggressive media. The chemical formula of vinyl-ester (the polyester modified by introducing the fragments of the corresponding bisphenol epoxy resin to the structure of the molecule) is presented in Figure 3.

The fillers included coarse and fine aggregates, the same as those used in ordinary Portland cement concrete, two commercially available quartz powders (produced by mechanical milling of quartz sand), and three by-products of the combustion of two fossil fuels—hard coal and lignite. The fillers included in particular:
conventional coarse aggregate: natural gravel of a fraction of 4/8 mm (marked G),conventional fine aggregate: standard sand (acc. to the EN-196-1 standard) or river sand of a fraction of 0/2 mm (marked S),conventional commercial microfiller: two quartz powders (marked Q_1_ and Q_2_),sustainable microfiller substitute: siliceous fly ash (conventional hard coal combustion by-product, marked FA_S_) and two fluidized fly ashes (hard coal FBC by-product, marked FA_F_, and lignite FBC by-product, marked FA_F2_).


Figure 4 presents the particle size distribution plots and SEM micrographs of all microfillers listed above. The particle size distribution measurements were done by the laser scattering method using the laser analyzer Horiba LA-300 (Kyoto, Japan). The method based on the Mie theory [23] involved passing laser beams through a 0.2% sodium polymetaphosphate solution containing microfiller particles (additionally dispersed by ultrasounds) and determining the particle size (in the range of 0.01–600 μm). The SEM micrographs were made with a Hitachi TM-1000 Scanning Electron Microscope (Tokyo, Japan) using a back scattered electrons (BSE) detector. The values of statistical parameters describing the fillers’ particle size distributions, as well as specific surface area (calculated from the distribution, making an assumption about the spherical shape of the particles) are given in Table 1.

The microfillers were characterized by different particle morphologies and grading. The lowest values of maximum particle size, D_max_ (67.52 µm and 58.95 µm), were registered in the case of quartz powders (Q_1_ and Q_2_). On the contrary, siliceous fly ash (FA_S_) contained the largest particles (size up to 200 µm), and its particle size distribution showed the highest values of the mode and median (4.96 µm and 25.30 µm, respectively). In the case of fly ashes from fluidized combustion (FA_F_ and FA_F2_), the mode was 0.25 µm, which was 20 times lower compared to siliceous fly ash, while the median was 0.26 µm or 0.27 µm—values two orders of magnitude lower than in the case of siliceous fly ash. It is worth noting that the grading of both fluidized fly ashes was described by a bimodal distribution. The abovementioned mode value applied to the increased number of particles smaller than 1 µm. A similar mode value (0.28 µm) was observed in the case of quartz powder Q_1_, which was why fly ashes from FBC seemed to be more similar to this conventional quartz filler in terms of grading. The fly ash originating from lignite fluidized bed combustion (FA_F2_) was slightly thicker and had a smaller specific surface area than the fly ash from hard coal fluidized bed combustion (FA_F_). Both, however, had a more developed specific surface area than quartz powders. Meanwhile, in terms of morphology, siliceous fly ash containing spherical grains was more similar to quartz powders, which contained angular grains resulting from mechanical milling of quartz.

The density was also determined for all microfillers. The test was performed in a Le Chatelier volumenometer (consisting of a flat-bottomed flask) according to the procedure described in the EN 1936 standard. The results of density are given in Table 1.

Polymer concrete-like composites prepared for testing contained vinyl-ester binder, conventional fine (S) and coarse (G) aggregate, and a mix of quartz powder (Q_1_ or Q_2_) and chosen fly ash (FA_S_, FA_F_, or FA_F2_). The quantitative compositions of tested composites are presented on Figure 5. The compositions were determined according to the statistical Box design (variant of the CCD design [24]) assuming three factors that were expressed as the mass ratios of the components. The first variable was the ratio of the amount of binder and the amount of basic aggregate (i.e., gravel and sand), B/(G + S), in the range 6.0–10.0. The second variable was the ratio of the amount of binder and the total amount of microfiller fraction (including quartz powder and fly ash), B/(FA + Q), in the range of 0.4–0.6. The third variable was the ratio of the amount of fly ash and the amount of microfiller fraction, FA/(FA + Q), in the range of 0.0–1.0. The use of such a design allowed obtaining composites of various quantitative composition, which were predestined for statistical analysis. The quantitative compositions were identical to the compositions of concretes tested in previously mentioned studies, so it was possible to compare the results.

Specimens tested in the presented research were the halves of prisms with dimensions of 40 × 40 × 160 mm^3^ remaining after the flexural test (there-point bending test). The presented compressive strength values were the average (of 4 or 2 results). Specimens were stored in the laboratory conditions for periods of 14 days (time of curing of the polymer concretes recommended in the EN 1542 standard), 9 years in the case of concretes with fly ashes remaining from hard coal combustion (FA_F_ and FA_S_), or 9.5 years in the case of concretes with fly ash remaining from lignite combustion (FA_F2_).

The change in the compressive strength in time was considered the measure of the long-term performance. It was legitimate to speak of “change” in compressive strength, because the specimens originated from the same prisms, and the results obtained for the halves of the same prisms were compared. Each time, one half of a prism was compressed after 14 days, and the other half was stored and destroyed after 9 or 9.5 years.

Before the destructive tests were carried out, the specimens were also examined in terms of the volumetric density. The density was determined according to the method described in the EN 12390-7 standard. The density was calculated on the basis of measurements of mass and volume obtained by water displacement (i.e., the method for determining the density of irregularly-shaped specimens; in this particular case, the abovementioned halves of prisms remaining after flexural test).

## 4. Results and discussion

### 4.1. Compressive Strength of Vinyl-ester Concretes with Fly Ash from Lignite Combustion

The results of compressive strength tests of concretes containing various contents of vinyl-ester binder and particular fillers, including quartz powder Q_2_ and fly ash of lignite FA_F2_ (in amounts as shown in the Figure 5), are given in the Figure 6. For an easier discussion, the composites were grouped in terms of the share of fly ash in the microfiller (values in the round brackets) and, generally, the amount of fly ash in the composite.

As in the case of vinyl-ester concretes containing fly ashes remaining from hard coal combustion, also in the case of analogous composites containing identical amounts of fly ash from lignite combustion, and in the case of a number of composites with even more diverse quantitative compositions, it was confirmed that the compressive strength of such concrete-like polymer composites significantly increased with time.

The increase in strength of the tested composites ranged from 9.5 MPa to even 27.5 MPa (on average by 16.6 MPa), which corresponded to percentage changes in the range of 9.1–28.8%. In the case of initially very weak concrete containing a small amount of vinyl-ester resin (10% by composite mass) and almost 80% of fluidized fly ash in the microfiller (Composition ID No. 8), which after 14 days of curing as characterized with compressive strength of 14.8 MPa, after 9.5 years, it turned out to be twice as strong (31.9 MPa).

There was no clear explanation as to how the development of compressive strength was progressing for all the composites of various quantitative compositions. However, analyzing the obtained data, one could recognize that vinyl-ester concretes of the same range of substitution quartz microfiller with fly ash (FA/M = 21% or FA/M = 50%) showed similar tendencies (with the test probability *p*-value = 0.95, α = 0.05) with a good correlation (R > 0.8): the more fly ash in the composite, the smaller the increase in strength over time (Figure 7).

This effect, however, could be easily explained. Fly ash FAF_2_ was much finer than quartz powder Q_2_, which it replaced (compare the appropriate median and mode values listed in Table 1), and therefore, the fly ash was a component with the finest grains. Thus, the more the fly ash was in the composite, the easier it filled the inter-grain voids of the other fillers’ fractions. This meant that the microstructure of concrete with a higher content of fly ash was tighter at the start and left less possibility for its physical densification of the structure in later times, so the increase in compressive strength over time should also be smaller.

Figure 8 presents the micrographs of two vinyl-ester concretes with different fluidized fly ash contents (21% and 79%) in microfiller fraction taken shortly after the specimens’ preparation (in 2010). The SEM micrographs were made with a Hitachi TM-1000 Scanning Electron Microscope using a BSE detector. Before making the micrographs, the specimens were impregnated in epoxy resin (in a vacuum chamber) and prepared by multi-stage grinding (in the presence of diamond suspensions) and polishing. Surface preparation was carried out using the Struers TegraPol-21 grinding-polishing machine and the Struers TegraDoser-5 diamond suspension dispensing kit.

The SEM micrographs showed that the microstructure of vinyl-ester concrete with a higher content of fly ash and lower content of quartz powder in the microfiller fraction (FA/M = 79%, Q/M = 21%) was well compacted, and very fine particles of fly ash were regularly distributed in the polymer phase. In the case of vinyl-ester concrete in which quartz powder predominated in the microfiller (FA/M = 21%, Q/M = 79%), one could see that between the large angular grains of quartz, there were spaces filled with a polymer less saturated with mineral filler.

### 4.2. Development of Compressive Strength of Vinyl-ester Concretes with Various Fly Ashes

The mechanical tests results obtained for concrete with fly ash of lignite FBC (Figure 6) clearly showed that in the case of such a filler, the increase of the compressive strength of polymer concrete proceeded for a longer time. However, these results did not answer the question about when there was a progressive increase in the compressive strength of vinyl-ester concretes with fly ashes and whether it ended before the test date (i.e., earlier than 9.5 years after the composites were made). To explore this issue, it is worth looking at the additional results recently obtained for identical composites (in qualitative and quantitative composition) as discussed in the article cited in the Introduction. For vinyl-ester concretes with fly ashes originating from hard coal combustion (specifically, Specimen ID No. 3 and No. 7 according to Figure 5), new compressive strength tests were carried out nine years after the specimens’ preparation, and the results were compared with the data obtained after 14 days and after seven years (data published in [20]). Table 2 contains such a data sheet. Moreover, the table contains data obtained for concrete with the same quantitative composition, but with fly ash remaining from lignite combustion.

In the case of the first series of composites (ID: No. 3 acc. to Figure 5) of identical quantitative composition differing only in the type of fly ash, a greater increase in strength (both after seven and nine years), was found when using siliceous fly ash FA_S_ (Δ_7_ = 9.6%, Δ_9_ = 27.4%) than in case of fly ash from fluidized hard coal combustion, FA_F_ (Δ_7_ = 3.4%, Δ_9_ = 14.2%). However, it should be noted that the strength values of composites determined after the same longer time were almost identical, both after seven years (103.2 MPa and 104.4 MPa) and after nine years (120.0 MPa and 115.3 MPa). An analogous concrete with FA_2_ fly ash from lignite combustion tested after 9.5 years was characterized with practically imperceptibly higher strength of 122.4 MPa.

In case of the second series of composites (Specimen ID No. 7 acc. to Figure 5), the compressive strength determined after nine or 9.5 years for composites with various fly ashes also adopted very close values (differences up to 2.5 MPa). This may indicate that a period of nine years was sufficient to obtain a full range of strength.

When it came to the period of seven years, it was difficult to formulate one conclusion. Again, the amount of the finest fraction—the fly ash—should be taken into account. For a series of composites containing less fly ash, i.e., 21% fly ash in the general microfiller fraction (5.3% of the total composite mass, Specimen ID No. 3), the difference between strength tested after seven and nine years was significant. Meanwhile, for concrete containing much fluidized fly ash, i.e., 79% of microfiller fraction (20.1% of the composite mass, Specimen ID No. 7), no increase in strength between seven and nine years was noted (the values were very close—110.8 MPa and 107.5 MPa). Thus, in the last case, seven years seemed to be sufficient time to achieve a full strength.

For each polymer concrete compressive strength test, the coefficient of variation (within-batch variation) was calculated. The CV ranged from 1.4–7.1% for composites containing less fly ash and 1.9–9.7% for concretes richer in fly ash. Such values could be considered satisfactory if they were assessed in the context of ordinary concrete with a similar strength, whereas given other composite material and specimens sizes, there are no particular guidelines for such an assessment [25,26].

One can expect that the numerous grains of the finest fillers, thus the smallest particles of microfiller fraction, could be the physical obstacles to the free formation of structures of polymerized chains of the binder; thus, the polymerization proceeded slower, and therefore, the polymer matrix strengthened for a longer time before the composite obtained its final properties. Moreover, the additional cross-linking of the polymer could have been initialized by calcium compounds, which were present in significant amounts in the by-products of fluidized combustion. This would explain why, in the case of a composite richer in fluidized fly ash, compressive strength stabilized faster.

### 4.3. Volumetric Density

Conclusions regarding the potential densification of the microstructure deduced on the basis of the compressive strength gain in time were confirmed by the results of the measurement of the density of composites. Table 3 contains data regarding the change in volumetric density over time of the abovementioned composites with fly ashes remaining after coal combustion (FA_S_ and FA_F_) at the age of 14 days and nine years. One can observe that in the case of composites in which more quartz powder (with a density of 2650 kg/m^3^) was replaced by lighter siliceous fly ash (with density of 2110 kg/m^3^) or fluidized fly ash (with a density of 2440 kg/m^3^), i.e., composites with a lower initial volumetric density, the increase in density was greater. 

## 5. Conclusions

The following conclusions emerged from the investigations of the long-term compressive strength of vinyl-ester concretes with various microfillers:The results confirmed that in the case of tested polymer concretes prepared with the use of commercially available vinyl-ester resin as a binder, a significant improvement of compressive strength after a period of several years was noted. This was evidence that vinyl-ester concretes were durable polymer composites when analyzed in terms of mechanical strength criterion.The increase in compressive strength was noted regardless of the characteristic or origin of the microfiller used, i.e., in the case of commonly used commercial quartz powders and when using by-products remaining after combustion of various fossil fuels in various installations (including fly ashes from FBC, whose impact on the long-term properties of polymer composites had not been recognized until this point).As shown, the increase in compressive strength continued for several years; however, the values of strength stabilized for different composites after different times, depending on the amount of microfiller the vinyl-ester binder was saturated with: the less fly ash, the longer the time until strength stabilization. Nonetheless, the obtained results indicated that the period of nine years was sufficient to obtain a full range of strength of tested PC. Moreover, in cases of composite rich in fluidized fly ash, the shorter period of seven years seemed to be sufficient.The morphology, particles size distribution, and density of the microfillers used affected microstructure and, as a consequence, the volumetric density of hardened composites: the more fly ash in the composite, the lower the initial volumetric density of PC, but the greater the increase in density with time. After nine years, the density of all tested concretes adopted had very similar values of ca. 2200 kg/m^3^.

To explain the phenomena, the physical effects of which were the subject of this paper, the author plans additional long-term research focused on changes in the internal structure and microstructure of polymer concretes for the various stages of their service-life.

## Figures and Tables

**Figure 1 materials-13-01207-f001:**
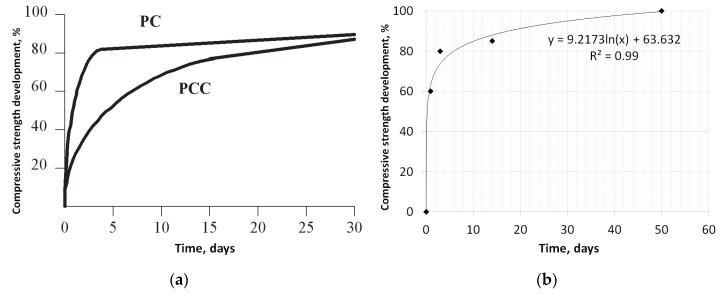
Development of the compressive strength of polymer composites expressed as the relative increase in time: (**a**) model of polymer concrete (PC) and polymer-cement concrete (PCC) developed by Czarnecki [2]; (**b**) recalculation of the first model using the natural logarithm function.

**Figure 2 materials-13-01207-f002:**
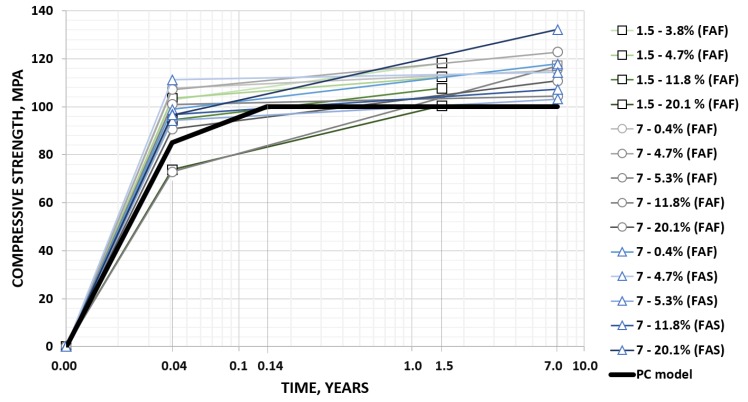
Development of the strength of polymer concretes with various mass contents (0.4–20.1%) of fly ash in a microfiller fraction in time; FA_F_, fly ash from fluidized bed combustion (FBC); FA_S_, siliceous fly ash from conventional combustion (chart based on data published by the author in [20] compared to the model calculated on the basis of [2]).

**Figure 3 materials-13-01207-f003:**
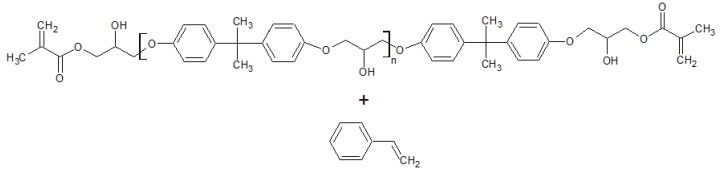
Vinyl-ester resin before cross-linking [22].

**Figure 4 materials-13-01207-f004:**
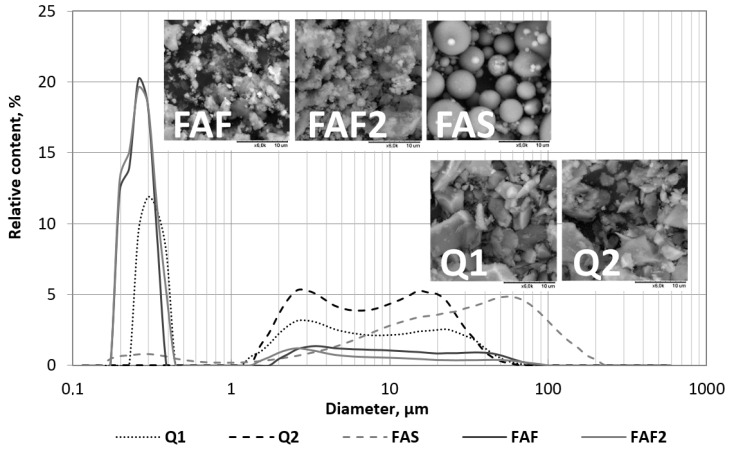
Particle size distribution plots and SEM micrographs (magnification: 6000×) of the considered fillers (Q_1_, Q_2_, quartz powders; FA_S_, siliceous fly ash; FA_F_, fluidized fly ash from hard coal combustion; FA_F2_, fluidized fly ash from lignite combustion).

**Figure 5 materials-13-01207-f005:**
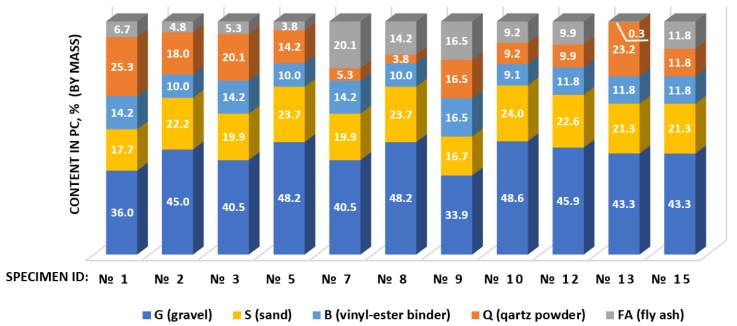
Vinyl-ester concretes compositions (% by mass) based on the statistical Box design (experimental CCD design of three factors expressed as the mass ratios of components: B/(G + S) in range of 6.0–10.0, B/(FA + Q) in the range of 0.4–0.6, and FA/(FA + Q) in the range of 0.0–1.0).

**Figure 6 materials-13-01207-f006:**
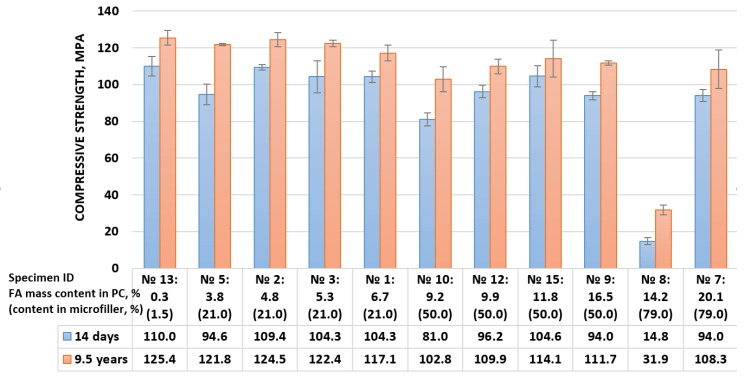
Compressive strength of vinyl-ester concretes of various mass contents of fly ash remaining from lignite FBC obtained after 14 days and 9.5 years; FA, fly ash; PC, polymer concrete.

**Figure 7 materials-13-01207-f007:**
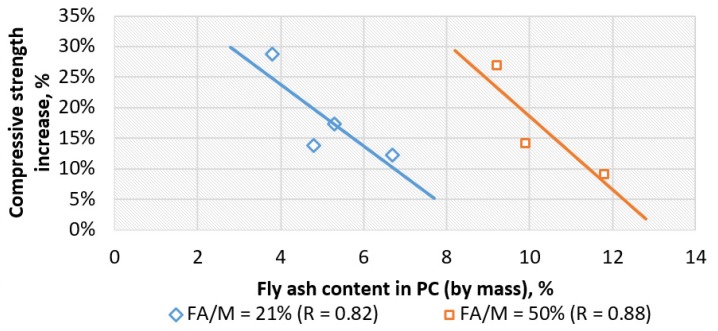
The increase in compressive strength of vinyl-ester concretes with various content of fly ash of lignite FBC registered after 9.5 years in relation to fly ash mass content; the tendencies (statistical significance level α = 0.05) were determined separately for concretes where 21% of the quartz microfiller was substituted by fly ash (FA/M = 21%) and concretes where half of the quartz microfiller was substituted by fly ash (FA/M = 50%).

**Figure 8 materials-13-01207-f008:**
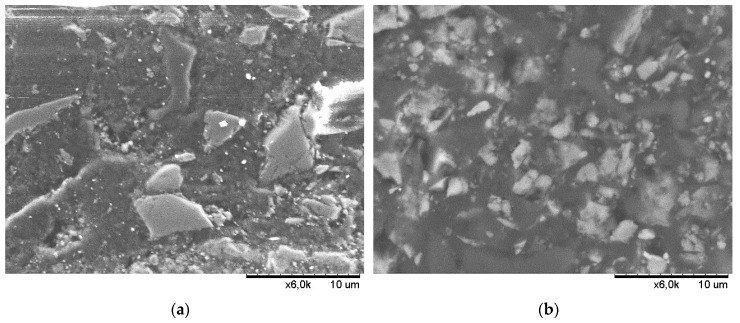
SEM micrographs (magnification: 6000×) of vinyl-ester composites with various contents of fly ash FA_F_: (**a**) composite with 21% content of fly ash in the microfiller (ID: No. 3 acc. to Figure 5); (**b**) composite with 79% content of fly ash in microfiller (ID: Np. 7 acc. to Figure 5)

**Table 1 materials-13-01207-t001:** Statistical parameters describing the particle size distribution and specific surface area (SSA) of fillers; Q_1_, Q_2_, quartz powders; FA_S_, siliceous fly ash; FA_F_, FA_F2_, fluidized fly ashes.

Raw Material	Quartz Sand	Quartz Sand	Coal	Coal	Lignite
Production	Grinding	Grinding	Conventional Combustion	FBC Combustion	FBC Combustion
Parameter	Filler Q_1_ [20]	Filler Q_2_	Filler FA_S_ [20]	Filler FA_F_ [20]	Filler FA_F2_
D _min_, µm	0.26	1.32	0.17	0.12	0.17
Mode, µm	0.28	2.45	4.96	0.25	0.25
Median, µm	2.44	7.18	25.30	0.27	0.26
D _max_, µm	67.52	58.95	200.00	77.34	101.46
SSA ^1^, m^2^/m^3^	8 701	5 857	18 294	14 825	12 215
Density, kg/m^3^	2650	2650	2110	2440	2550

^1^ SSA was calculated from the particle size distribution, making an assumption about the spherical shape.

**Table 2 materials-13-01207-t002:** Compressive strength of chosen vinyl-ester composites (ID: No. 3 and No. 7 acc. to Figure 5) with various types of fly ash (FA_S_, siliceous fly ash; FA_F_, fly ash from hard coal FBC; FA_F2_, fly ash from lignite FBC) obtained after 14 days, 7 years, and 9 years or after 14 days and 9.5 years.

Fly Ash	ID	FA/M, %	FA/PC, kg/kg	Age: 14 Days		Age: 7 Years		Age: 9 or 9.5 Years		Δ_7_/Δ_9/9.5_, %
				f_c_, MPa	CV, %	f_c_, MPa	CV, %	f_c_, MPa	CV, %	
FA_S_	№ 3	21	5.3	94.2	3.3	103.2	0.0	120.0	3.3	9.6/27.4
FA_F_				101.0	1.2	104.4	0.5	115.3	7.1	3.4/14.2
FA_F2_				104.3	8.4	-	-	122.4 ^1^	1.4	- /17.4
FA_S_	№ 7	79	20.1	96.6	3.6	-	-	105.0	4.7	- /8.7
FA_F_				90.7	4.8	110.8	0.0	107.5	1.9	22.2/18.5
FA_F2_				94.0	3.5	-	-	105.2 ^1^	9.7	- /11.9

^1^ Specimens with fly ash from lignite FBC tested after 9.5 years.

**Table 3 materials-13-01207-t003:** Density of composites with various fly ashes (FA_S_, siliceous fly ash; FA_F_, fluidized fly ash) determined after 14 days and 9 years.

Fly Ash	ID	FA/M, %	FA/PC, kg/kg	Age: 14 Days		Age: 9 Years		Gain	
				D, kg/m^3^	CV, %	D, kg/m^3^	CV, %	ΔD, kg/m^3^	ΔD, %
FA_S_	№ 3	21.0	5.3	2149	0.7	2201	3.7	52	2.4
FA_F_				2162	0.24	2185	0.5	23	1.1
FA_S_	№ 7	79.0	20.1	2112	2.3	2201	2.2	89	4.2
FA_F_				2102	1.4	2188	0.8	86	4.1

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
