# Peer review of "Long-Term Compressive Strength of Polymer Concrete-like Composites with Various Fillers"

_materials, 2020, doi:10.3390/ma13051207_

Round 1
Reviewer 1 Report
This re-submission of the manuscript presents an improved research work. A long-term study of the mechanical strengths of polymer-concrete samples has been carried out. Alternative fine fillers, coal by-products, have been tested.
The writing of the manuscript has been improved, although I still think that Introduction is rather complex and sometimes difficult to read. An improvement of this section should be undertaken.
In addition, I am not convinced about the use of Figures in the Introduction. Maybe the author could add another section as Background.
Line 205-206: please, remove the statement about the long-term durability, since the experimental measurement did not assess the performance of the specimens under aggressive conditions (just the mechanical strength).
In Figure 6, please, add error bars.
Fig. 7: Correlation studies should include the associated probability of the statistical test.
Fig. 8b is rather blurred: can the author provide a sharper micrograph?
Line 305: please, write properly coefficient of variation.
Section devoted to Volumetric density should be section 3.3, revise the wrong numbering.
Lines 346-355 are not conclusions but speculative arguments (largely probable). They should be removed from Conclusions and could be added to the Discussion section.
Author Response
Dear Reviewer,
One more time thank you for valuable remarks. I have made all indicated corrections. Pleas find below my comments to the particular remarks.
Kind Regards,
Joanna J. Sokołowska, Ph.D.
The Introduction as recommended has been divided into two sections („Introduction” and „Genesis of research”). The Figures are included in the second section. Also the text of „Introduction” has been revised one more time.
As recommended the sentence in lines 205-206 (at this moment 206-207) has been changed. The term „durability” has been changed to „performance”.
In Figure 6 the error bars has been added.
As for Fig. 7, the probability of the statistical test details have been added in the figure caption as well as in the text in lines 239-240.
Fig. 8b – unfortunately I do not have sharper micrograph.
Line 305 – typing mistake has been corrected.
Section devoted to Volumetric density has been numbered as 3.3.
Lines 346-355 – The indicated text has been removed from Conclusions and moved to the Discussion section (Section 3.2).
Reviewer 2 Report
Authors improved significantly the manuscript according to my comments.
Please revise “coefficient of vitiation” at line 305.
Author Response
Dear Reviewer,
Thank you for paying so much attention to my paper and for the valuable comments.
Yours Sincerely,
Author, Joanna J. Sokołowska
Reviewer 3 Report
The authors have enriched the manuscript and clarified a significant number of points. The soundness of the paper and quality of presentation of methods and results has also been improved. In my opinion the paper is now publishable.
Author Response
Dear Reviewer,
Thank you for paying so much attention to my paper and for the valuable comments.
Yours Sincerely,
Author, Joanna J. Sokołowska
This manuscript is a resubmission of an earlier submission. The following is a list of the peer review reports and author responses from that submission.
Round 1
Reviewer 1 Report
Please see attached file.

Reviewer 2 Report
The manuscript deals with the durability of building composites with polymer matrix, mainly in terms of strength variation during time, looking at different compositions and considering a quite long-time span. This makes the results significant and useful for the research community, however there are some suggestions to improve the manuscript before potential publication.
Please support with a reference or with better clarification, why “The change in the compressive strength in time was considered as the measure of the long-term durability.” At line 159
Please comment also on the expected strength variability in the same batch of composite and its comparability with the strength increases with time; in other word, how is it expected that the specimens tested after years would have a different strength compared with the specimens tested after 14 days even if they were tested after 14 days? However I see that it is a systematic increase, but it would be interesting to explain also this aspect in the text.
At the end of page 6 it seems that “physical compaction later” is the reason for the increase of strength, while earlier it seems that also reaction during time or its delay can be a reason for strength increases over time. Can you please clarify the expected sources of increase of strength during time, in the text?
Looking at table 2 it seems that the increase between 14 days and 7 years is lower than the increase between 7 and 9 years, both in absolute and percentage terms (quite different from figure 1b and the logarithmic interpolation). It seems that there is no explanation for this.
Please provide some reasons or references to support the change in volumetric density over time (section 3.2).
Reviewer 3 Report
This manuscript presents a long-term study of the mechanical strengths of some polymer-concrete samples. The specimens were prepared with alternative fine fillers, such as coal by-products.
The writing of the manuscript is not appropriate: the author mentions too many times results previously published, making difficult the understanding of the paper and forcing the reader to go in depth into too many different previous papers (such as those in 146-151). The main points, such as those related to the samples’ composition and the most relevant previous results, should be included in the current manuscript.
The paper is eminently descriptive and does not go into the main reasons behind the performance of the samples tested after long-term. The most serious lack is the absence of a rigorous microstructural analysis, including SEM observations and/or pore size distribution measurements. To ascertain the role of the different fillers, this analysis seems to be imperative. The author only suggests this analysis in the Conclusions (lines 258-260).
The title does not match the content of the manuscript. Instead of durability (which involves the exposure to more or less “aggressive” conditions), it would be better to mention “Long-term mechanical strengths…”. The author did not undertake an assessment of the durability, either under common climatic conditions or under accelerated ageing conditions (climatic chamber, SO2 chamber, freezing-thawing cycles, salt fog chamber…).
Introduction is rather complex and sometimes difficult to read. The author should improve this part, focusing on the new aspects tackled in the current manuscript. The previous findings should be reported in a summarized way, avoiding the use of Figures in the Introduction.
Figure 5, 6, Table 2, 3, : revise the decimal points
Figure 5. Revise the composition of FA in sample 13.
Fig. 7: Correlation studies should include the associated probability of the statistical test.
In its current form the manuscript cannot be accepted. A new submission should be prepared, including a new collection of experiments analyzing the microstructure of the samples and providing a solid scientific support. The writing should be entirely modified. The main previous findings should be summarized and included in the text (avoiding repetition of data, of course).